# Long-Term Prognosis after Coronary Artery Bypass Grafting: The Impact of Arterial Stiffness and Multifocal Atherosclerosis

**DOI:** 10.3390/jcm11154585

**Published:** 2022-08-05

**Authors:** Alexey N. Sumin, Anna V. Shcheglova, Sergey V. Ivanov, Olga L. Barbarash

**Affiliations:** Federal State Budgetary Scientific Institution “Research Institute for Complex Issues of Cardiovascular Disease”, Sosnoviy Blvd., 6, 650002 Kemerovo, Russia

**Keywords:** cardio-ankle vascular index, multifocal atherosclerosis, coronary artery disease, coronary artery bypass grafting, long-term prognosis

## Abstract

The aim of the study was to study the effect of arterial stiffness and multifocal atherosclerosis on the 10-year prognosis of patients after coronary artery bypass grafting. Methods. Patients with coronary artery disease (*n* = 274) who underwent coronary artery bypass grafting (CABG), in whom cardio-ankle vascular index (CAVI) was assessed using the VaSera VS-1000 device and the presence of peripheral atherosclerosis in Doppler ultrasound. Groups were distinguished with normal CAVI (<9.0, *n* = 163) and pathological CAVI (≥9.0, *n* = 111). To assess the prognosis, coronary and non-coronary death, myocardial infarction, acute cerebrovascular accident/transient ischemic attack, repeated CABG, percutaneous coronary intervention, carotid endarterectomy, peripheral arterial surgery, pacemaker implantation were analyzed. Results. During the observation period, mortality was 27.7%. A fatal outcome from all causes was in 37 (22.7%) patients in the group with normal CAVI and in 39 (35.14%) in the group with pathological CAVI (*p* = 0.023). Death from cardiac causes was more common in the group with CAVI ≥ 9.0—in 25 cases (22.52%) than in the group with CAVI < 9.0—in 19 (11.6%, *p* = 0.016). The combined endpoint in patients with pathological CAVI was detected in 66 (59.46%) cases, with normal CAVI values—in 76 (46.63%) cases (*p* = 0.03). The presence of diabetes mellitus, multifocal atherosclerosis (*p* = 0.004), pathological CAVI (*p* = 0.063), and male gender were independent predictors of death at 10-year follow-up after CABG. The presence of multifocal atherosclerosis and pathological CAVI during the preoperative examination of patients were independent predictors of the combined endpoint development. Findings. Patients with coronary artery disease with pathological CAVI before CABG were more likely to experience adverse events and death in the long-term follow-up than patients with normal CAVI. Further studies are needed to investigate the possibility of correcting pathological CAVI after CABG after secondary prevention and the possible impact of this correction on prognosis.

## 1. Introduction

Long-term results after coronary artery bypass grafting depend not only on the completeness of revascularization, on the initial severity of coronary and myocardial lesions, but also on comorbidity (diabetes mellitus, arterial hypertension, diseases of the lungs, kidneys, etc.) [1,2]. In addition, of importance is the state of the vascular system as a whole, which can also affect the course of the disease after coronary artery bypass grafting (CABG). It has now been shown that multifocal atherosclerosis (that is, damage to several arterial basins) adversely affects the prognosis after coronary bypass surgery [3,4,5]. These studies included patients with clinical evidence of extra-cardiac vascular disease (history of transient ischemia/stroke, intermittent claudication, decreased ABI < 0.9, need for arterial revascularization) [3,5]. Although some studies have taken into account subclinical manifestations of extra-cardiac vascular disease (for example, arterial stenosis of more than 50% that does not require surgical treatment [4]), the impact of such changes on prognosis was not separately assessed.

There is less data on the influence of such a factor as arterial stiffness on the condition of patients after CABG [6]. It should be noted that the prognostic value of arterial stiffness in population studies is known [7], in patients with coronary artery disease [8,9], including patients with myocardial infarction [10,11], and after percutaneous coronary intervention (PCI) [12]. However, the assessment of arterial stiffness in coronary artery bypass grafting has been poorly studied to date. Early studies attempted to use brachial pulse pressure as a marker of arterial stiffness in CABG [13,14], and it was shown that increased pulse pressure is associated with poor prognosis in CABG. However, at the same time, it was recognized by experts that the pulse pressure on the brachial artery could not be an accurate marker of arterial stiffness, since it differs from the central pressure measured in the aorta [15,16]. Therefore, to assess the relationship between arterial stiffness and the results of CABG, instrumental methods for its assessment are required, based on measuring the speed of propagation of a pulse wave along the arterial bed [7]. A limited number of studies dealt with the effect of increased arterial stiffness on the immediate results of CABG: on the development of acute renal failure [17,18], and the total number of postoperative complications [19,20]. Of course, there is evidence in the literature that acute renal failure after CABG adversely affects long-term prognosis [21], but a direct assessment of the effect of increased arterial stiffness on long-term prognosis has not been performed. Only in our group’s study has it been shown that increased arterial stiffness is associated with a poor prognosis at mid-term follow-up [22], but whether this effect persists at longer follow-up remains an open question.

Another important issue concerns methods for assessing arterial stiffness. The study of arterial stiffness using pulse wave velocity assessment has significant limitations (requirements for operator qualifications, technical difficulties in conducting the study, and difficulties in standardization) [7]. In addition, the dependence on the level of blood pressure makes it difficult to use this indicator for dynamic monitoring (it is not certain whether its change is associated with the lability of the blood pressure level, or whether the state of the vascular wall has actually changed). Therefore, a new cardio-ankle vascular index (CAVI) has been proposed, which reflects the stiffness of the arterial tree from the beginning of the aorta to the ankle [23]. It is now recognized that CAVI has several unique properties [24]. Firstly, ease of measurement (with BP cuffs placed on both arms and ankles and a microphone on the chest, without the need for sensors in the neck or groin) and independence from the operator. Second, CAVI reflects the stiffness of the entire aorta (including the ascending segment), femoral, popliteal, and tibial arteries, and measures the increase in arterial stiffness occurring from end diastole to end systole. Thirdly, CAVI is less influenced by blood pressure during measurement compared to PWV, since CAVI is based on the stiffness parameter β, which reflects the degree of pressure-volume relationship [24]. In studies predominantly on Asian populations, this index showed an association with both cardiovascular risk factors and prognosis for cardiovascular disease [25]. The main disadvantage of this method, at the present time, is that most of the studies have been performed in Asia and it remains unclear to what extent the obtained data can be extended to other populations. Therefore, studies of this index in other countries and ethnic groups are extremely topical [7]. It can also be noted that an increase in arterial stiffness and the development of atherosclerosis of the arterial wall may reflect various pathophysiological processes [26], and there is evidence of an increase in prognostic value when jointly assessing arterial stiffness and the ankle-brachial index in myocardial infarction [11]. Therefore, it seems interesting to study the complex effect of these factors on the prognosis after CABG. This was the basis for the present study, the purpose of which was to study the effect of arterial stiffness and multifocal atherosclerosis on the 10-year prognosis of patients after coronary artery bypass grafting.

## 2. Methods

### 2.1. Participants

We studied CAD patients who underwent CABG in the cardio-vascular surgery department of the Research Institute for Complex Issues of Cardiovascular Diseases between March 2011 and March 2012. A detailed characterization of the studied cohort of patients is presented in our previous article [22]. We did not include in the analysis patients with diseases that could affect the values of CAVI. These included recent acute coronary syndrome, valvular heart disease, low ejection fraction, low ankle-brachial index (≤0.9), atrial fibrillation, and pacemaker. The study protocol was approved by the Local Ethics Committee of the Federal State Budgetary Institution “Research Institute for Complex Issues of Cardiovascular Diseases” (Protocol No. 20110216). All patients signed an informed consent prior to enrollment in the study.

### 2.2. Baseline Data

Baseline patient data were obtained from the electronic database of the institute’s CABG registry. For each patient, the following data were collected: severity of angina pectoris and heart failure, the number of affected coronary arteries, previous myocardial infarction and coronary interventions, operations on peripheral arteries, risk factors, comorbid conditions, and preoperative drug therapy. Also, patients underwent transthoracic echocardiography with an assessment of the structural parameters of the heart, as well as assessed the state of peripheral arteries in duplex ultrasonography and the arterial stiffness using CAVI. We also took into account intraoperative parameters: EUROSCORE scale values, the use of cardiopulmonary bypass and its duration, the bypass grafts number, and combined operations.

### 2.3. Measurement of CAVI

We assessed arterial stiffness by determining the cardio-ankle vascular index (CAVI) using the VaSera VS-1000 device (Fukuda Denshi, Tokyo, Japan) according to the previously described method [22]. This index is calculated automatically by the device on both lower extremities, it does not depend on the qualification of the operator, on the level of blood pressure, and is highly reproducible. At the same time, the device allows assessing the ankle-brachial index, with ABI values ≤ 0.9; patients were not included in the study, since in such cases the CAVI values were underestimated. If the value of CAVI ≥ 9.0 on at least one side, the index was considered pathological. CAVI values of 9.0 as a threshold for distinguishing between normal and pathological values were proposed by the authors of the proposed index, therefore, such threshold values of CAVI are often used in studies [25,27]. We also followed this approach in the present study.

### 2.4. Duplex Ultrasonography Assessment

The study was carried out by experienced sonographers on an expert class ultrasound device ‘Vivid 7 Dimension’ (General Electric, Boston, MA, USA), and a 7.5 MHz linear-array transducer following a standard protocol [22]. We measured the thickness of the intima-media complex and assessed the state of the vessel: patency, and the structure of the plaque (in Doppler and B-mode). Mild stenosis was determined by the degree of narrowing of the carotid artery in 30–49% and moderate and severe stenosis—in 50% and more.

We also performed duplex ultrasonography of the arteries of the lower extremities with an assessment of their condition from the common femoral to the artery of the foot. The common femoral and anterior tibial arteries were visualized with the patient in the supine position, and the popliteal, peroneal, and posterior tibial arteries were visualized in the lateral position. The severity of arterial stenosis was classified in a similar way as in the evaluation of the carotid arteries.

The presence of multifocal atherosclerosis was established if, along with the existing lesion of the coronary arteries, there were arterial stenoses in at least one of the peripheral arterial basins (stenoses of 30% or more or 50% or more were distinguished). The choice of these cut-off values for the characterization of subclinical stenoses was made empirically, based on previous studies in our clinic.

### 2.5. Echocardiographic Examination

Transthoracic echocardiography was performed in the preoperative period on an expert-class Vivid-7 Dimension machine (General Electric, USA) by experienced ultrasound doctors who did not know the CAVI values, in accordance with current recommendations [28]. Among the structural indicators, the dimensions and volumes of the left ventricle in systole and diastole, and the maximum transverse size of the left atrium (LA) in diastole were evaluated. Left ventricular ejection fraction (LVEF) was calculated using the Simpson method. In the right parts of the heart, the dimensions of the right atrium (RA) and the dimensions of the right ventricle (RV) were assessed [22].

### 2.6. Follow-Up

A cardiologist or general practitioner in their residence place observed patients included in the study for ten years. To obtain information about patients after 10 years, we contacted the patient or his relatives by phone and clarified the presence of any events during this time, the presence of symptoms, and the therapy he received. As a result, information about the state of health was obtained from 274 (70%) patients (Figure 1). Two groups were formed depending on preoperative CAVI: group I CAVI < 9.0 (*n* = 163), and group II CAVI ≥ 9.0 (*n* = 111). We analyzed the following events: death (cardiac, non-cardiac, unexplained), non-fatal myocardial infarction, non-fatal stroke or transient ischemic attack, re-interventions on the coronary arteries (re-CABG, percutaneous coronary intervention), and other cardiovascular operations (carotid endarterectomy, peripheral arteries surgery, and pacemaker implantation). The primary endpoint included death from all causes. Secondary endpoints included combined endpoint (all of the above adverse outcomes), and combined endpoint + recurrent angina.

### 2.7. Statistical Analysis

Standard STATISTICA 8.0 (Dell Software, Inc., Round Rock, TX, USA) and SPSS 17.0 (IBM, Armonk, NY, USA) software were used for statistical analyses. A case-control study design was used to retrospectively analyze the data of groups with pathological (≥9.0), and normal (<9.0) CAVI. Qualitative values were presented in absolute numbers (*n*) and percentage (%), and comparisons between the groups were performed using χ^2^ tests. The normality of the quantitative data distribution was verified using the Kolmogorov-Smirnov test. For a distribution other than normal, all quantitative variables were presented as the median, low, and upper quartiles (ME [LQ, UQ]). When comparing quantitative data in two groups, the Mann–Whitney test was used. To assess the relationship of binary traits (death from all causes; combined endpoint; and combined endpoint + recurrent angina) with preoperative and perioperative variables binary logistic regression analysis (Forward LR method) was used. The level of critical significance (*p*) was taken as being equal to 0.05.

## 3. Results

### 3.1. Baseline Characteristics

Table 1 presents the preoperative clinical and anamnestic characteristics of the examined patients in groups with normal and pathological CAVI. The proportion of men was 76%. Patients with normal CAVI were younger than patients with abnormal CAVI (*p* < 0.001). In the group of patients with a pathological index, anamnestic cardiovascular risk factors, clinical manifestations of CAD, and chronic heart failure prevailed, and a higher score on the EuroScore II scale, which characterizes the surgery risk, was noted. All patients underwent a series of laboratory and instrumental examinations, according to the results of which the groups were comparable (Table 2). Only multifocal atherosclerosis with arterial stenosis ≥ 30% before CABG was detected in every second patient in the group with pathological CAVI (*p* = 0.0008). The prevalence of multifocal atherosclerosis with stenosis ≥ 50% was almost one-third of patients with pathological CAVI (*p* = 0.003).

According to the data on coronary angiography, there were no significant intergroup differences. However, in patients with a pathological CAVI, multivessel coronary artery disease was more often visualized. In addition, no differences were found in the groups in terms of intraoperative characteristics (Table 3).

### 3.2. Long-Time Outcomes after CABG in Patients with Pathological and Normal CAVI

When evaluating the long-term prognosis, one cannot fail to note in both groups a significant decrease in cardiovascular risk factors, such as BMI (up to 24.4 [21.6; 26.6] kg/m^2^ and 23.7 [21.5; 26.6 kg/m^2^] in groups with normal and pathological CAVI, *p* = 0.7) and smoking (up to 17.8% and 8.0%, respectively, *p* = 0.06). The resumption of the angina clinic was registered equally often in both groups (33.1% and 32.4%, *p* = 0.9). The total number of coronary angiography performed during the observation period did not differ in the study groups (30.7% and 32.4%, *p* = 0.75) and was comparable to the clinical manifestations of angina pectoris. The frequency of visits to a cardiologist in the long-term period was 1.0 ± 3.0 times a year. Analysis of the received drug therapy showed not very high adherence to the prescribed treatment (67.2% in all patients). At the same time, the groups did not differ in the intake of optimal drug therapy, although the frequency of drug intake prevailed in the group with normal CAVI (Figure 2).

During the observation period, mortality was 27.7%. The lethal outcome from all causes was in 76 cases: 37 (22.7%) in the normal CAVI group and 39 (35.14%) in the pathological CAVI group (*p* = 0.023) (Figure 3). A detailed analysis found that death from cardiac causes was significantly more common in the group with CAVI ≥ 9.0—25 cases (22.52%) than in the group with CAVI <9.0—19 cases (11.6%, *p* = 0.016) (Figure 4, Appendix A). The analysis of cardiovascular events revealed a higher frequency of non-fatal MI (*p* = 0.047) and PCI (*p* = 0.057) in the group with normal CAVI than in the pathological CAVI group. The groups did not differ in the frequency of stroke, or surgical interventions on the peripheral arteries. In general, combined endpoint (death, non-fatal MI, stroke/TIA, PCI, pulmonary embolism, re-CABG, and peripheral artery surgeries) were registered in 142 patients (51.8%). In patients with pathological CAVI, combined endpoint was more often observed (in 66 cases, 59.46%), compared with patients with normal CAVI (in 76 cases, 46.63%, *p* = 0.03). Similar differences were found between groups when additionally accounting for recurrent angina (Figure 3).

### 3.3. Predictors of the Unfavorable Long-Time Outcomes after CABG (Binary Logistic Regression Analysis, Forward LR Method)

The presence of diabetes mellitus, multifocal atherosclerosis (*p* = 0.004), and male gender were independent predictors of death at 10-year follow-up after CABG (Table 4). Pathological CAVI (*p* = 0.063) was also represented in a binary logistic regression equation, the logistic regression model was statistically significant, χ^2^(5) = 24.0, *p* < 0.001. The model explained 15.7% (Nagelkerke R2) of the variance in death and correctly classified 76.5% of cases. The presence of multifocal atherosclerosis and pathological CAVI during the preoperative examination of patients were independent predictors of the combined endpoint development. For this model statistical significance was χ^2^(2) = 11.3, *p* = 0.004, Nagelkerke R2 value was 0.07, and model correctly classified 58.5% of cases. When taking into account both the combined endpoint and recurrent angina, diabetes mellitus and multifocal atherosclerosis were included in the predictive model. In this model χ^2^(2) was 12.9 (*p* = 0.002), Nagelkerke R2 value—0.08, and model correctly classified 66.8% of cases.

## 4. Discussion

The main result obtained in this study is that long-term results after coronary artery bypass grafting, largely depend on the initial state of the vascular wall, not only on the prevalence of the atherosclerotic process but also on increased arterial stiffness. We have shown for the first time that even the presence of a subclinical lesion of extra-cardiac vascular disease (stenosis of more than 30% in at least one of the basins) is associated with a long-term prognosis in patients with coronary artery disease after CABG. In addition, for the first time, an independent association of the CAVI index with the development of a combined endpoint in this category of patients was shown during a 10-year follow-up.

Surprisingly, to date, there are not many studies evaluating the effect of arterial stiffness on the CABG results. In the 2000s, it was shown that elevated pulse pressure in the preoperative period was associated with poor outcomes after CABG. Therefore, in a multicenter study, it was shown that in patients with pulse pressure >80 mm Hg, the incidence of cerebral events and/or death from neurological complications nearly doubled, as did elevation of the congestive heart failure incidence by 52%, and cardiac death by almost 100% immediately after CABG [13]. In addition, Nikolov et al. [14] showed that an increase in pulse pressure was a significant predictor of a decrease in long-term survival after CABG. An increase in pulse pressure is associated with an increase in aortic stiffness and can serve as a surrogate marker for an increase in arterial wall stiffness. In a 2018 study [29], these data were confirmed, that baseline pulse pressure was significantly associated with postoperative acute kidney injury after CABG surgery. However, it has recently been shown that elevated preoperative pulse pressure was associated with only modest increases in postoperative troponin-T concentrations, but not with postoperative cardiovascular complications or in-hospital mortality in patients undergoing CABG [30]. Therefore, there was a need to assess arterial stiffness with a more accurate method.

It has now been shown that arterial stiffness assessment using pulse wave velocity (PWV) before elective CABG was an independent predictor of the development of cardiac surgical acute kidney injury (CSA-AKI) [17,18], as well as the development of the incidence of stroke and delirium (composite neurologic outcome) [17]. When studying another method of revascularizatio—percutaneous coronary intervention—increased PWV was associated with a higher incidence of stent restenosis during follow-up within six months after the intervention [31]. Another method for assessing arterial stiffness—CAVI—was studied in our studies. It was previously shown that pathological CAVI is associated with a higher incidence of perioperative strokes and deaths [19], as well as more likely to develop cardiovascular complications and cardiovascular death within a subsequent five-year follow-up [22]. The present study confirmed the previously obtained data with a longer follow-up. In this study, in contrast to the previous study [22], it was also shown that not only increased arterial stiffness but also the presence of diabetes mellitus and manifestations of subclinical multifocal atherosclerosis affected the long-term prognosis after CABG. In addition, the fact of maintaining the effect on the prognosis of patients after CABG of the initial increased vascular stiffness at a 10-year follow-up makes researchers pay more attention to this parameter in the postoperative management of patients.

Multifocal atherosclerosis is a significant factor that worsens the prognosis after CABG. This has been shown in different cohorts of patients, for example, in the various age groups [4], during on-pump and off-pump CABG [5]. This complements the findings of Aboyans V et al., who showed that subclinical lower extremity artery disease (ABI < 0.85 without symptoms of intermittent claudication) was an independent predictor of adverse events at 5-year follow-up [32]. The difference between our study and the above is the inclusion of patients with even less pronounced lesions of the lower extremities arteries (patients with an ABI ≤ 0.9 were excluded). However, and in this cohort of patients, the presence of subclinical multifocal atherosclerosis turned out to be an independent predictor of adverse outcomes during a 10-year follow-up of patients.

What is the clinical significance of this study? Our data show that in patients with coronary artery disease before CABG, it is advisable to identify patients with initial manifestations of multifocal atherosclerosis since such patients have an increased risk of adverse outcomes during long-term follow-up. Perhaps even more practical will be the assessment of arterial stiffness. On the one hand, there is an opinion that the assessment of arterial stiffness is a simple non-invasive method that can potentially be used for risk stratification after elective cardiac surgery (for example, PWV to predict the development of acute kidney injury [18]). In this regard, the use of the CAVI index is more convenient from a practical point of view—measurement technology is simpler and standardized, and CAVI does not depend on the blood pressure level (the latter is especially convenient for dynamic monitoring) [33]. On the other hand, the independent predictive value of CAVI at long-term follow-up is also useful from the clinicians’ point of view. Increased arterial stiffness is a hallmark of the aging process and is manifested by the development of arteriosclerosis, in contrast to atherosclerosis with the development of local plaques. Although the presence of peripheral arterial disease is included in the long-term risk assessment models after CABG [27,34,35], the assessment of arterial stiffness using CAVI provides additional opportunities. Since the CAVI index reflects the influence of various risk factors on the vascular wall [23], it is convenient to use it to assess the effectiveness of secondary prevention measures [8]. Moreover, experts believe that arterial stiffness is an important therapeutic goal to improve the prognosis in CAD patients [8]. Our data that an elevated CAVI index is independently associated with long-term prognosis after CABG suggest that in this category of patients, the effect on arterial stiffness may be the goal of secondary prevention. However, this assumption needs to be confirmed in further studies. In patients with stable CAD with conservative treatment, it has been shown that the absence of a decrease in CAVI after six months was associated with an unfavorable prognosis [36]. The traditional fight against risk factors (smoking, dyslipidemia, diabetes mellitus, arterial hypertension [2]) is manifested, among other things, by an improvement in the elasticity of the vascular wall, but it is not clear how pronounced this can be in patients after CABG; such studies have not been conducted. The question also remains open—is it possible to influence the long-term prognosis after CABG if it is possible to achieve a decrease in CAVI in the course of therapeutic and prophylactic interventions.

## 5. Study Limitations

We did not include some patients with comorbid pathology (rhythm disturbances, low ejection fraction, low ABI values, valvular pathology) in order to be able to correctly assess CAVI. Therefore, the predictive value of CAVI is limited to this sample of patients and cannot be extended to all CAD patients who underwent CABG.

In addition, we did not include patients with severe atherosclerosis of the peripheral arteries of the lower extremities, which could affect the relative contribution of multisite arterial disease and the CAVI index to the long-term prognosis after CABG.

The sample size is relatively small and does not enable sufficient adjustment to potential confounders, therefore, there may be doubts that the presented data indicate CAVI as an independent prognostic marker, and it can only be a confounding factor. It is possible to consider our data as preliminary, which should be confirmed in a study with a larger sample or in a multicenter study. However, obtaining such results may be only in 10 years, even if the study is now planned. In any case, in our opinion, these studies deserve publication in order to raise the question of the scientific community about the confirmation of our data.

Patients with coronary artery disease and clinical manifestations of multivascular disease, often need to be treated more intensively, but we did not purposefully monitor therapy in such patients. Patients were treated by doctors at their place of residence in accordance with current recommendations. However, the cohort of examined patients included predominantly patients with subclinical manifestations of multifocal atherosclerosis, for which there are no well-established recommendations. The inclusion of received therapy in multiple regression models also did not reveal an independent effect on prognosis.

We did not evaluate CAVI over time with ongoing therapy and secondary prevention, which does not allow us to understand whether it is possible to reduce CAVI in patients after CABG and, if possible, whether such a decrease affects the prognosis in patients after CABG. It seems that further research in this direction should help answer these questions.

## 6. Conclusions

Patients with coronary artery disease with pathological CAVI prior to CABG had more frequent adverse events and death at long-term follow-up than patients with normal CAVI. The presence of subclinical multifocal atherosclerosis and pathological CAVI were independent predictors of the development of the combined endpoint. During a 10-year follow-up along with such predictors of death as female sex, diabetes mellitus, and multifocal atherosclerosis, the borderline significance of the pathological CAVI was also noted. Further studies are needed to investigate the possibility of correction of pathological CAVI after CABG following secondary prevention and the possible impact of this correction on prognosis.

## Figures and Tables

**Figure 1 jcm-11-04585-f001:**
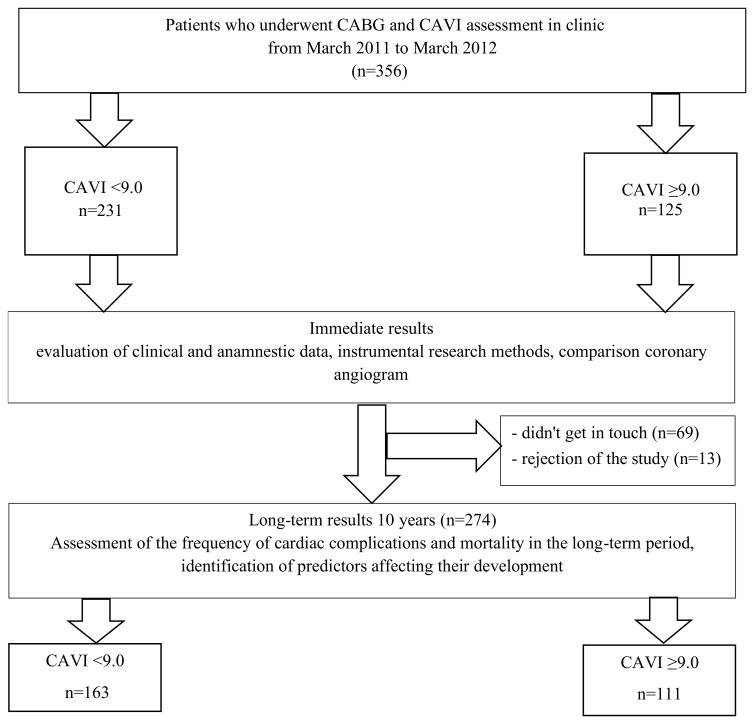
Flowchart of patient selection. CABG—coronary artery bypass surgery, CAVI—cardio-ankle vascular index.

**Figure 2 jcm-11-04585-f002:**
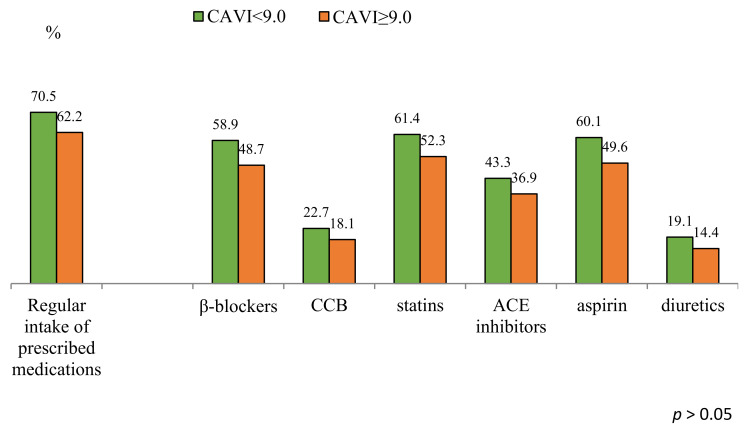
Therapy of patients 10 years after CABG with normal and pathological CAVI. Notes: CCB—calcium channel blockers, ACE—angiotensin converting enzyme; CABG—coronary artery bypass grafting; CAVI—cardio-ankle vascular index.

**Figure 3 jcm-11-04585-f003:**
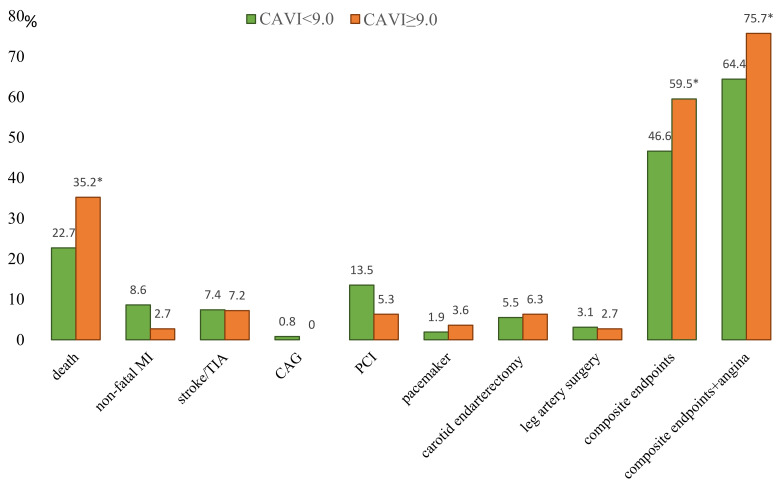
Structure of complications in the long-term period after CABG. Notes: MI—myocardial infarction; TIA—transient ischemic attack; PCI—percutaneous coronary intervention; CAG—coronary angiography; CABG—coronary artery bypass grafting; CAVI—cardio-ankle vascular index. * *p* < 0.05 compared with normal CAVI group.

**Figure 4 jcm-11-04585-f004:**
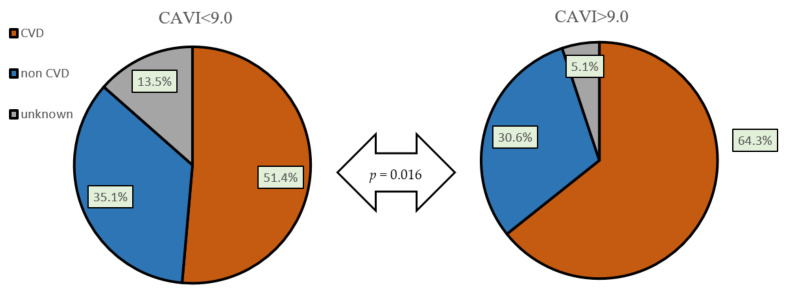
Structure of mortality in the long-term period. Notes: CVD—cardiovascular disease; CAVI—cardio-ankle vascular index.

**Table 1 jcm-11-04585-t001:** Baseline characteristics in groups with pathological and normal CAVI before coronary artery bypass surgery (*n* = 274).

Variables	Group 1 CAVI < 9.0 *n* = 163		Group 2 CAVI ≥ 9.0 *n* = 111		*p*
	No. of Events	%	No. of Events	%	
Age, Me [LQ; UQ] years	57.0 [52.0; 61.0]		63.0 [55.0; 69.0]		<0.001
BMI, Me [LQ; UQ] kg/m^2^	28.02 [24.5; 32.1]		28.37 [26.6; 30.5]		0.57
EuroScore, Me [LQ; UQ]	2.0 [1.0; 3.0]		3.0 [2.0; 4.0]		0.005
Male gender	129	79.14	80	72.07	0.17
Myocardial infarction history	106	65.03	64	57.66	0.21
Hypertension	133	81.6	104	93.69	0.004
Stroke history	8	4.91	9	8.11	0.28
Transitory ischemic attack	1	0.61	2	1.8	0.35
Diabetes mellitus	19	11.66	23	20.72	0.04
PCI history	13	7.98	9	8.11	0.96
CABG history	1	0.61	1	0.9	0.78
Carotid endarterectomy	2	1.23	3	2.7	0.37
Current smoking	53	32.52	28	25.23	0.19
No angina	29	17.9	27	24.32	0.19
Angina I functional class	8	4.94	3	2.7	0.35
Angina II functional class	55	33.95	29	26.13	0.16
Angina III functional class	68	41.98	48	43.24	0.83
Angina IV functional class	2	1.23	4	3.6	0.18
Heart failure, class NYHA I	109	66.87	60	54.1	0.032
Heart failure, class NYHA II	42	25.77	41	39.94	0.048
Heart failure, class NYHA III	4	2.45	4	43.6	0.57
Preoperative Drug Therapy					
β-blockers	107	65.6	75	67.6	0.840
Calcium channel blockers	57	35.0	33	30.0	0.438
Statins	81	49.6	60	54.1	0.557
Angiotensin-converting enzyme inhibitors	78	47.8	53	47.7	0.707
aspirin	108	66.2	79	71.2	0.285
Oral antihyperglycemic therapy	9	5.5	10	9.0	0.264
Insulin therapy	5	3.1	5	4.5	0.534

Notes: CAVI—cardio-ankle vascular index; LQ—low quartile; UQ—upper quartile; CABG—coronary artery bypass grafting; NYHA—New York Heart Association; PCI—percutaneous coronary intervention; BMI—body mass index.

**Table 2 jcm-11-04585-t002:** The instrumental and laboratory parameters in groups with pathological and normal CAVI before coronary artery bypass surgery.

Variables	Group 1 CAVI < 9.0 *n* = 163	Group 2 CAVI ≥ 9.0 *n* = 111	*p*
Laboratory data			
Total cholesterol (mmol/L)	4.9 [4.1; 5.9]	5.05 [4.2; 6.1]	0.27
HDL cholesterol (mmol/L)	0.96 [0.82; 1.165]	1.01 [0.84; 1.2]	0.33
LDL cholesterol (mmol/L)	2.91 [2.28; 3.79]	2.96 [2.2; 3.74]	0.37
Triglycerides (mmol/L)	1.76 [1.37; 2.43]	1.66 [1.22; 2.3]	0.45
Creatinine (µmol/L)	83.0 [70.0; 100.0]	82.5 [72.0; 102.0]	0.93
Glucose (mmol/L)	5.55 [5.1; 6.4]	5.45 [5.0; 6.3]	0.67
Preoperative echocardiogram			
LV EDD, cm	5.5 [5.2; 6.0]	5.5 [5.2; 6.1]	0.98
LV ESD, cm	3.8 [3.4; 4.6]	3.7 [3.5; 4.6]	0.68
LV EDV, mL	153.0 [135.0; 180.0]	147.0 [135.0; 188.0]	0.95
LV ESV, mL	62.0 [47.0; 87.0]	60.0 [49.0; 84.0]	0.75
LA, cm	4.1 [3.8; 4.4]	4.3 [3.9; 4.6]	0.007
RV, cm	1.8 [1.8; 1.9]	1.8 [1.8; 1.8]	0.24
RA, cm	4.0 [3.8; 4.5]	4.25 [3.9; 4.5]	0.17
LV EF, (%)	60.0 [51.0; 64.0]	60.0 [52.0; 63.0]	0.32
CIMT, mm	1.1 [1.0; 1.2]	1.1 [1.0; 1.2]	0.32
Atherosclerosis lesions of arterial basins			
Carotid artery stenoses ≥ 30%, *n*(%)	27 (16.56)	20 (18.02)	0.75
Carotid artery stenoses ≥ 50%, *n*(%)	18 (11.04)	21 (18.92)	0.06
Carotid artery stenoses both sides ≥ 30%, *n*(%)	19 (11.66)	19 (17.21)	0.19
Multifocal atherosclerosis ≥ 30%, *n*(%)	55 (33.74)	55 (49.55)	0.0008
Multifocal atherosclerosis ≥ 50%, *n*(%)	21 (12.88)	30 (27.03)	0.003

Note: Continuous data are presented as median (lower quartile, upper quartile). Abbreviations: CAVI—cardio-ankle vascular index; LDL-low-density lipoproteins; CIMT—carotid intima-media thickness; HDL—high-density lipoproteins; LV EDD— left ventricular end-diastolic dimension; LV ESD—left ventricular end-systolic dimension; LV EDV—left ventricular end-diastolic volume; LV ESV—left ventricular end-systolic volume; LA—left atrium; RV—right ventricle, RA—right atrium.

**Table 3 jcm-11-04585-t003:** Surgical procedure and coronary angiography in groups with pathological and normal CAVI.

Variables	Group 1 CAVI < 9.0 *n* = 163		Group 2 CAVI ≥ 9.0 *n* = 111		*p*
	No. of Events	%	No. of Events	%	
1-coronary artery disease	28	17.18	20	18.02	0.85
2-coronary artery disease	56	34.36	34	30.63	0.51
3-coronary artery disease	71	43.56	50	45.05	0.81
LMCA ≥ 50%	36	22.09	19	18.02	0.41
Cardiopulmonary bypass	140	85.89	93	83.78	0.63
Ventriculoplasty	9	5.52	5	4.5	0.7
Thrombectomy	6	3.68	2	1.8	0.36
Carotid endarterectomy.	3	1.84	2	1.8	0.98
Radiofrequency ablation	2	1.23	3	2.7	0.37
Bypass graft number Me [LQ; UQ]	3.0 [2.0; 3.0]		3.0 [2.0; 3.0]		0.71
Cardiopulmonary bypass duration, Me [LQ; UQ] min	98.0 [79.0; 110.0]		94.0 [79.5; 107.5]		0.59
Operation duration Me [LQ; UQ] min	240.0 [198.0; 300.0]		240.0 [204.0; 270.0]		0.82

Abbreviations: CAVI—cardio-ankle vascular index; LMCA—left main coronary artery; LQ—low quartile; UQ—upper quartile.

**Table 4 jcm-11-04585-t004:** Predictors of the unfavorable long-time outcomes after CABG (binary logistic regression analysis, Forward LR method).

Death						
	B	S.E.	Wald	df	Sig.	Exp (B)
Male gender	1.032	0.448	5.313	1	0.021	2.807
Diabetes mellitus	0.838	0.426	3.874	1	0.049	2.312
PAD	−1.737	0.813	4.567	1	0.033	0.176
Multifocal atherosclerosis ≥ 30%	1.030	0.360	8.169	1	0.004	2.800
CAVI ≥ 9.0	0.646	0.348	3.445	1	0.063	1.907
Constant	−2.716	0.498	29.724	1	0.000	0.066
**Combined endpoint**						
Multifocal atherosclerosis ≥ 30%	0.705	0.292	5.843	1	0.016	2.025
CAVI ≥ 9.0	0.589	0.286	4.250	1	0.039	1.802
Constant	−0.480	0.204	5.543	1	0.019	0.619
**Combined endpoint + recurrent angina**						
Diabetes mellitus	1.296	0.509	6.481	1	0.011	3.656
Multifocal atherosclerosis ≥ 30%	0.681	0.321	4.498	1	0.034	1.975
Constant	0.302	0.185	2.663	1	0.103	1.353

Notes: CABG—coronary artery bypass grafting; LR—Likelihood Ratio; PAD—peripheral artery disease; CAVI—cardio-ankle vascular index.

## Data Availability

The datasets used and/or analyzed during the current study available from the corresponding author on reasonable request.

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
