# Peer review of "Long-Term Prognosis after Coronary Artery Bypass Grafting: The Impact of Arterial Stiffness and Multifocal Atherosclerosis"

_jcm, 2022, doi:10.3390/jcm11154585_

Round 1

Reviewer 1 Report

The reviewer approves of the improvement of the content. However, there are areas where the content has not been sufficiently improved. Have you checked it properly?

- There is something wrong with the contents of the lower part of Table 2. Specifically, the relevant numbers and P-values.

Author Response

- There is something wrong with the contents of the lower part of Table 2. Specifically, the relevant Thank you for your comment. Indeed, this is an error, we have corrected it.

Thank you for your comment. Indeed, this is an error, we have corrected it.

Reviewer 2 Report

To the authors,

   In this study, Sumin AN et al. reported the 10-years follow-up analysis of patients who underwent coronary artery bypass grafting (CABG) for the purpose to reveal prognostic significance of baseline preoperative CAVI and degree of atherosclerotic change assessed by ultrasonography.  CAVI is a marker of vascular stiffness and an arteriosclerosis indicator of thoracic, abdominal, common iliac, femoral, and tibial arteries independent of arterial blood pressure. Accordingly, the authors found that patients with pathological CAVI (>9.0) before CABG surgery were more likely to have adverse events and death in the long-term follow-up than in patients with normal CAVI (<9.0) group. This study is longer time follow-up analysis came from the same cohort of their prior published 5-years follow-up study in patient who underwent CABG (Sumin AN, Glob Heart. 2021). Insights from detailed 10-years follow-up data in patients who underwent CABG is impressive, worthy to present. However, at the same time, there are several critical concerns to be mentioned as follows.

Major comments

1.     (Table 2); What is the units of “Atherosclerosis lesions of arterial basins” including Carotid artery stenoses and multifocal atherosclerosis? Are these the number of patients? P values of these columns seems not adequate (i.e. 20, 21, 19…… these cannot be exact P values).

2.     (Table 3); What is the variable of “OnepaIiNR c NK”? Why Thrombectomy, Carotid endarterectomy, and Radiofrequency ablation were included in the variables list? Were these procedures simultaneously conducted with CABG surgery? 

3.     (Figure 2); What is “systematic use”? Please explain the details.

4.     The reason why 0.9 was selected as a threshold of CAVI for grouping in this study seemed to be unclear. Reasonable explanation needs to be added.

5.     (Figure 3); CAG should not be involved in the list, cannot be counted as a complication. What is  “OCHOBHOH” in the Y axis?

6.     According to the Figure 3, the incidence of MI was less in CAVI≥9.0 group. However, Figure 4 clearly showed greater percentage of cardiovascular death in CAVI≥9.0 group. How the authors explain the discrepancy? Details of cause of death including CVD and non-CVD in both groups need to be mentioned. 

7.     (Table 4); Before conducting binary logistic regression analysis, univariate analyses to explore significant variables in patients who underwent unfavorable long-time outcomes needs to be shown. Then the variables which involved in the logistic regression analysis should be selected with fair reasons. Indeed, older age was highly associated with CAVI ≥9.0 in Table1. Did the authors include “age” as a variable in the logistic regression analysis? 

8.     In the table, “Sex” should be “Male gender”. As a predictor for death event, CAVI ≥9.0 failed to show statistical significance (i.e. Sig=0.063). It cannot be described as an independent predictor of death at 10-years follow-up after PCI in the main document (Page 11, line 262-263). 

9.     The authors have already conducted 5-years follow-up study regarding CAVI in the same cohort (Sumin AN, Glob Heart. 2021). Just extending follow-up period is not enough to show. What are the new findings of the current 10-years follow-up data compared with their prior 5-years study? This point must be clearly discussed in depth. Without showing new aspect, the novelty of this study seems to be quite limited.

Minor comment

1.     (Page 10, line 251) What does “CT” stand for? 

Author Response

First of all, we would like to thank the reviewer for the work done in evaluating our manuscript and for useful comments. Eliminating the noted shortcomings has helped us, we hope, to improve our manuscript.

Major comments

  1. (Table 2); What is the units of “Atherosclerosis lesions of arterial basins” including Carotid artery stenoses and multifocal atherosclerosis? Are these the number of patients? P values of these columns seems not adequate (i.e. 20, 21, 19…… these cannot be exact P values).

 Thank you for your comment. Indeed, this is an error, we have corrected it.

  1. (Table 3); What is the variable of “OnepaIiNR c NK”? Why Thrombectomy, Carotid endarterectomy, and Radiofrequency ablation were included in the variables list? Were these procedures simultaneously conducted with CABG surgery? 

Sorry, this is an error again, right: “Cardiopulmonary bypass”, we have corrected it. Quite right, Thrombectomy, Carotid endarterectomy, and Radiofrequency ablation were included in the variables list because these procedures simultaneously conducted with CABG surgery

  1. (Figure 2); What is “systematic use”? Please explain the details.

 We have refined the variable in the figure 2. New version - "Regular intake of prescribed medications"

  1. The reason why 0.9 was selected as a threshold of CAVI for grouping in this study seemed to be unclear. Reasonable explanation needs to be added.

CAVI values of 9.0 as a threshold for distinguishing between normal and pathological values were proposed by the authors of the proposed index, therefore, such threshold values of CAVI are often used in studies [25,34]. We also followed this approach in the present study.

  1. (Figure 3); CAG should not be involved in the list, cannot be counted as a complication. What is  “OCHOBHOH” in the Y axis?

Indeed, formally CAG cannot be considered a complication. However, use of CAG is usually caused by some reason (for example, an increase in angina attacks), so we still decided to take this indicator into account. However, isolated CAG without subsequent revascularization was rare in our sample (0.8% in the group with normal CAVI) and clearly had no effect on the results of the study. The technical error in Figure 3 has been corrected.

  1. According to the Figure 3, the incidence of MI was less in CAVI≥9.0 group. However, Figure 4 clearly showed greater percentage of cardiovascular death in CAVI≥9.0 group. How the authors explain the discrepancy? Details of cause of death including CVD and non-CVD in both groups need to be mentioned. 

Thank you for your comment. Indeed, we made an inaccuracy in Figure 3 - "Non-fatal MI" was correct. Non-fatal MI was more common in the group with normal CAVI, but in the CAVI≥9.0 group, myocardial infarction was more often accompanied by a fatal outcome. We provide details of cause of death, including CVD and non-CVD, in both groups in a supplementary table.

  1. (Table 4); Before conducting binary logistic regression analysis, univariate analyses to explore significant variables in patients who underwent unfavorable long-time outcomes needs to be shown. Then the variables which involved in the logistic regression analysis should be selected with fair reasons. Indeed, older age was highly associated with CAVI ≥9.0 in Table1. Did the authors include “age” as a variable in the logistic regression analysis? 

When conducting binary logistic regression analysis, we included most of the parameters that had significant or close to significant differences in tables 1, 2 and 3. Since we compiled binary logistic regression models for three different dependent variables (death from all causes; combined endpoint; and combined endpoint + recurrent angina), we did not include data from univariate analyses due to for the bulkiness of the obtained tables. Naturally, "age" was included as a variable in the logistic regression analysis, in all three cases it was not statistically significant (p=0.672; p=0.168 and p=0.168). We explained this fact that it was not age itself that affected long-term outcomes after CABG, but the severity of vascular damage and the presence of concomitant pathology.

  1. In the table, “Sex” should be “Male gender”. As a predictor for death event, CAVI ≥9.0 failed to show statistical significance (i.e. Sig=0.063). It cannot be described as an independent predictor of death at 10-years follow-up after PCI in the main document (Page 11, line 262-263). 

 Thank you for your comment, we have made the necessary changes to the table and text.

  1. The authors have already conducted 5-years follow-up study regarding CAVI in the same cohort (Sumin AN, Glob Heart. 2021). Just extending follow-up period is not enough to show. What are the new findings of the current 10-years follow-up data compared with their prior 5-years study? This point must be clearly discussed in depth. Without showing new aspect, the novelty of this study seems to be quite limited.

We made additions to the Discussion section:

“Also in this study, in contrast to the previous study [22], it was shown that not only in-creased arterial stiffness, but also the presence of diabetes mellitus and manifestations of subclinical multifocal atherosclerosis affected the long-term prognosis after CABG. In addition, the fact of maintaining the effect on the prognosis of patients after CABG of the initial increased vascular stiffness at a 10-year follow-up makes researchers pay more attention to this parameter in the postoperative management of patients.”

Minor comment

  1. (Page 10, line 251) What does “CT” stand for? 

Thanks for the note, this is a mistake, right - "combined endpoint"

This manuscript is a resubmission of an earlier submission. The following is a list of the peer review reports and author responses from that submission.

Round 1

Reviewer 1 Report

This manuscript shows the predictive ability of CAVI for a long-term prognosis of 10 years after CABG. There are several points that need to be revised.

Major:

-         Did your database include dialysis patients as a patient background affecting arterial stiffness?

-         Could excluding cases affect the primary outcome?

-         The reviewer thinks Figures 1 and 2 may be misplaced.

-         The authors need to show details of baseline drug treatment that affect prognosis.

-         This is very important. How did the authors correct the prognostic impact of the therapeutic agents, including diabetes drugs?

-         The reviewer thinks the figures require legends.

-          

Minor:

-         There is something wrong with the contents of the lower part of Table 2 (Carotid artery stenosis>30% ...).

-         Inconsistency of commas and points in P-values in Table 3.

-         Some of the text in Figures 2 and 4 are not printed.

-         Inconsistent order of CVD and non-CVD within Figure 4.

Author Response

First of all, we would like to thank the reviewer for the work done in reviewing our manuscript and for useful comments which will enable us to improve our manuscript. For questions and comments, we can answer the following: 

Major:

-         Did your database include dialysis patients as a patient background affecting arterial stiffness?

Our database did not include dialysis patients. In general, CABG surgery for this category of patients is extremely rare in our clinic.

-         Could excluding cases affect the primary outcome?

Patients with conditions that could make it difficult to evaluate CAVI (condition after acute coronary syndrome, ABI less than 0.9, valvular pathology, low left ventricular ejection fraction) were initially excluded from the study. It is clear that these factors could adversely affect the prognosis of patients with CABG, but since this patients were excluded from the analysis, this could not affect the primary result.

-         The reviewer thinks Figures 1 and 2 may be misplaced.

Thank you for your comment. Indeed, figures 1 and 2 are mixed up, we have eliminated this technical error.

-         The authors need to show details of baseline drug treatment that affect prognosis.

Indeed, this is our omission, the article contains only data on therapy 10 years after the operation. We entered data on baseline drug treatment in Table 1.

-         This is very important. How did the authors correct the prognostic impact of the therapeutic agents, including diabetes drugs?

Indeed, this question is important. The principles of drug therapy were comparable in the studied groups. During CABG the patients received the usual therapy prescribed by the cardiologist of the Department of Cardiovascular Surgery. Table 1 shows data on such therapy in the studied groups, it did not differ. After the operation, patients were observed on an outpatient basis at the place of residence either by a cardiologist or by a general practitioner. Correction of therapy was carried out by these specialists. As shown in Figure 2 at the time of survey after 10 years, the groups also did not differ in the treatment received. Univariate analysis did not reveal the effect of the therapy received on the studied endpoints, therefore, the drugs received by patients were not included in the binary logistic regression equation.

-         The reviewer thinks the figures require legends.

We corrected the Figures 1 and 2, including legends and notes.

Minor:

-         There is something wrong with the contents of the lower part of Table 2 (Carotid artery stenosis>30% ...).

We once again carefully looked at the data in Table 2. Probably, they are not presented very clearly. It is clear that the presence of atherosclerosis of any peripheral arteries in the presence of coronary atherosclerosis suggests the presence of multifocal atherosclerosis. However, the lines containing information about carotid artery stenoses namely indicate the presence of a concomitant lesion of the carotid arteries. In the lines where the term Multifocal atherosclerosis is used, we are talking about the defeat of any of the peripheral pools (that is, it can be not only carotid stenosis, but also stenosis of the arteries of the lower extremities)

-         Inconsistency of commas and points in P-values in Table 3.

We have made corrections to the text of the table

-         Some of the text in Figures 2 and 4 are not printed.

We have made corrections to the Figures

-         Inconsistent order of CVD and non-CVD within Figure 4

We have made corrections to the Figure 4

Reviewer 2 Report

Dear Sir/Madam,

I had the opportunity to act as a reviewer on the recent submission by Sumin et al. to the Journal of Clinical Medicine.

The authors present interesting original research regarding the patients with coronary artery disease with normal versus pathological cardio-ankle vascular index before coronary artery bypass grafting. The main finding of the study was that these patients were more likely to experience adverse events and death in the long-term follow-up than in patients with normal cardio-ankle vascular index.

The results are interesting; however, I recommend checking spelling and grammar.

Furthermore, some major issues need to be addressed:

  1. I fail to find in the manuscript some details regarding sample size calculation. What are the comments of the authors on this matter?
  2. Regarding the measurement of the cardio-ankle vascular index: who performed the measurements and what qualification did these persons have? Are these reproducible? Can the authors provide an interobserver variability?
  3. How do the authors explain the higher rate of PCI and MI in the group with CAVI<9?
  4. I recommend correcting table 4 and transforming in order to make it reader-friendly. The statistics presented in this table are hard to follow.

Minor issues:

1-     In the abstract please define the abbreviations CAVI and CABG when first used.

Best regards,

Author Response

First of all, we would like to thank the reviewer for the work done in reviewing our manuscript and for useful comments which will enable us to improve our manuscript. For questions and comments, we can answer the following:

  1. I fail to find in the manuscript some details regarding sample size calculation. What are the comments of the authors on this matter?

We agree with the reviewer that the calculation of the sample size is important when studying the effect of the type of treatment used. However, our study design was non-interventional and we followed two cohorts of patients over a long period of time. As far as we know, in such cases, the calculation of the sample size is usually not carried out. With regard to the CAVI index, one can cite as an example the studies of Murakami K, et al (The Role of Cardio- Ankle Vascular Index as a Predictor of Mortality in Patients on Maintenance Hemodialysis. Vasc Health Risk Management. 2021, doi: 10.2147/VHRM.S339769) and Kirigaya J, et al. (Impact of Cardio- Ankle Vascular Index on Long-Term Outcome in Patients with Acute Coronary Syndrome. J Atheroscler Thromb. 2020, doi: 10.5551/jat.51409). In these studies, the authors also did not calculate sample sizes.

  1. Regarding the measurement of the cardio-ankle vascular index: who performed the measurements and what qualification did these persons have? Are these reproducible? Can the authors provide an interobserver variability?

The CAVI measurement was carried out by the same employee. This index was calculated automatically using the program of the VaSera device. We did not evaluate the reproducibility of measuring this index since this was done in previously published papers (e.g. Endes S, et al. Reproducibility of oscillometrically measured arterial stiffness indices: Results of the SAPALDIA 3 cohort study Scand J Clin Lab Invest 2015 Apr;75(2):170-6 doi: 10.3109/00365513.2014.993692)

  1. How do the authors explain the higher rate of PCI and MI in the group with CAVI<9?

We also drew attention to this fact. For ourselves, we explained it in this way. Since there were more deaths in the group of patients with pathological CAVI, it seems that myocardial infarctions were more severe in them and led to death. At the same time, patients with normal CAVI suffered less severe infarcts, respectively, in this group there were more non-fatal myocardial infarctions and PCI.

  1. I recommend correcting table 4 and transforming in order to make it reader-friendly. The statistics presented in this table are hard to follow.

Thanks for the suggestion, we have corrected the table by reducing the preliminary steps in the logistic regression analysis.

Minor issues:

  • In the abstract please define the abbreviations CAVI and CABG when first used.

We have made corrections to the text of the resume

Round 2

Reviewer 2 Report

Dear Sir/Madam,

Thank you for reviewing the manuscript and addressing the mentioned issues. These were adequately answered. Therefore, the manuscript seems suitable for publishing in the present form.

Best regards